# Targeting Pyruvate Kinase M2 and Lactate Dehydrogenase A Is an Effective Combination Strategy for the Treatment of Pancreatic Cancer

**DOI:** 10.3390/cancers11091372

**Published:** 2019-09-16

**Authors:** Goran Hamid Mohammad, Vessela Vassileva, Pilar Acedo, Steven W. M. Olde Damink, Massimo Malago, Dipok Kumar Dhar, Stephen P. Pereira

**Affiliations:** 1Institute for Liver and Digestive Health, Royal Free Hospital Campus, University College London, London NW3 2QG, UK; 2Komar Research Center, Komar University of Science and Technology, Sulaimani 46001, Iraq; 3Department of Surgery and Cancer, Imperial Centre for Translational and Experimental Medicine, Imperial College London, London W12 0UQ, UK; 4Department of Surgery, Maastricht University Medical Center & Nutrim School for Nutrition, Toxicology and Metabolism, Maastricht University, 6200 MD Maastricht, The Netherlands; 5Hepato-pancreatic-biliary and Liver Transplantation Surgery, Royal Free Hospital Campus, University College London, London NW3 2QG, UK; 6King Faisal Specialist Hospital and Research Center, Comparative Medicine Department and Organ Transplantation Center, Riyadh 11211, Saudi Arabia

**Keywords:** pancreatic cancer, pyruvate kinase M2, lactate dehydrogenase A, glycolytic enzymes, combination therapy

## Abstract

Reprogrammed glucose metabolism is one of the hallmarks of cancer, and increased expression of key glycolytic enzymes, such as pyruvate kinase M2 (PKM2) and lactate dehydrogenase A (LDHA), has been associated with poor prognosis in various malignancies. Targeting these enzymes could attenuate aerobic glycolysis and inhibit tumor proliferation. We investigated whether the PKM2 activator, TEPP-46, and the LDHA inhibitor, FX-11, can be combined to inhibit in vitro and in vivo tumor growth in preclinical models of pancreatic cancer. We assessed PKM2 and LDHA expression, enzyme activity, and cell proliferation rate after treatment with TEPP-46, FX-11, or a combination of both. Efficacy was validated in vivo by evaluating tumor growth, PK and LDHA activity in plasma and tumors, and PKM2, LDHA, and Ki-67 expression in tumor tissues following treatment. Dual therapy synergistically inhibited pancreatic cancer cell proliferation and significantly delayed tumor growth in vivo without apparent toxicity. Treatment with TEPP-46 and FX-11 resulted in increased PK and reduced LDHA enzyme activity in plasma and tumor tissues and decreased PKM2 and LDHA expression in tumors, which was reflected by a decrease in tumor volume and proliferation. The targeting of glycolytic enzymes such as PKM2 and LDHA represents a promising therapeutic approach for the treatment of pancreatic cancer.

## 1. Introduction

Pancreatic cancer is one of the most aggressive and lethal malignancies, leaving patients with a 5-year survival rate of approximately 10% [1,2]. Lack of early symptoms and detection leads to most patients being diagnosed with advanced inoperable disease and a median survival of less than a year. Current treatment regimens offer limited survival benefit and substantial toxicities; therefore, more effective therapeutic strategies need to be developed [3].

In recent years, the importance of metabolic alterations during malignant transformation has gained increased interest, and regulation of cancer cell metabolism is being investigated as a potential therapeutic intervention. Tumor cells adopt a unique metabolic phenotype, consuming high amounts of glucose and producing lactate through catalytic enzyme-mediated processes, even in the presence of adequate levels of oxygen, which is known as the Warburg effect or aerobic glycolysis [3,4]. Pyruvate kinase (PK) and lactate dehydrogenase A (LDHA) are two crucial glycolytic enzymes that facilitate these processes, conferring a growth advantage for tumor cells [5,6,7]. 

PK is the rate-limiting enzyme in the last step of the glycolytic pathway, catalyzing the conversion of phosphoenolpyruvate (PEP) to pyruvate. There are four PK isoforms—L, R, M1, and M2, expressed in the liver, erythrocytes, adult skeletal muscle and brain, and proliferating cells, respectively [5,7]. In these tissues, catalytically active PKM2 is a tetramer, but in tumor cells, PKM2 is present as a dimer and is unable to convert PEP to pyruvate [8,9]. As a result, there is an accumulation of glycolytic intermediates, which can be used for the biosynthesis of lipids, nucleic acids, and proteins, required by rapidly proliferating cells. Upregulated expression of PKM2 has been reported in multiple malignancies, including pancreatic cancer, and is associated with cell proliferation, survival, migration, invasion, and angiogenesis [7,10,11]. Several small molecule activators, which induce the tetramerization of PKM2 (activation), have been developed and have shown promising results in blocking glycolysis and inhibiting tumor cell proliferation [12,13,14], including TEPP-46, 6-((3-aminophenyl)methyl)-4-methyl-2-methylsulfinylthieno[3,4]pyrrolo[1,3-d]pyridazin-5-one [13,14,15]. TEPP-46 is a potent and selective activator of PKM2 (inducing PKM2 tetramerization) with no significant effects on PKL, PKR, and PKM1 isoforms. Treatment with TEPP-46 has shown altered metabolism in cultured cells and antitumor activity in preclinical models of lung and colorectal cancer [13,16,17].

LDHA is another crucial glycolytic enzyme that converts pyruvate to lactate and oxidizes the reduced form of nicotinamide adenine dinucleotide (NADH) to NAD^+^ to sustain glycolysis [18,19,20]. LDHA is predominantly expressed in muscle tissue and at low levels in other tissues. High LDHA expression has been reported in a number of malignancies, including pancreatic cancer, and has been associated with tumor development, invasion, and metastasis [21,22,23]. Targeting of LDHA with siRNA or small molecule inhibitors has been reported to increase oxygen consumption and reactive oxygen species production, reduce glucose uptake and lactate production, and decrease tumor cell growth [21,24,25,26]. FX-11, 3-dihydroxy-6-methyl-7-(phenylmethyl)-4-propylnaphthalene-1-carboxylic acid, is a selective, reversible, NADH competitive inhibitor of LDHA that has shown antitumor activity in lymphoma and pancreatic cancer xenografts [24,27,28]. Moreover, inhibition of LDHA in combination with gemcitabine has shown synergistic cytotoxic activity in pancreatic cancer cells [29]. A multivariate analysis identified serum LDH levels as a significant prognostic predictor and a statistical trend for overall survival of patients with advanced pancreatic cancer after gemcitabine-based chemotherapy [30]. Similar results were observed by Faloppi et al. in advanced pancreatic cancer patients receiving sorafenib [31]. Patients with low LDH serum levels treated with sorafenib showed an advantage in progression-free survival and in overall survival.

PKM2 and LDHA seem to synergistically catalyze the glycolytic process to promote oncogenic metabolic reprogramming and appear to play a crucial role during the initiation and progression of pancreatic cancer [32,33,34]. Indeed, we have shown that increased expression of both PKM2 and LDHA can contribute to the initiation and progression of pancreatic cancer and is associated with poor prognosis [35]. Furthermore, a recently published study confirmed that PKM2 and LDHA also play a key role in pancreatic cancer-associated fibroblasts (CAFs) metabolism. PKM2 and LDHA protein expression in CAFs was higher than that in normal fibroblast [36].

Therefore, targeting these glycolytic enzymes could be a potential therapeutic strategy that can attenuate aerobic glycolysis and inhibit tumor cell proliferation. We evaluated this concept by combining the PKM2 activator, TEPP-46, with the LDHA inhibitor, FX-11, and assessed efficacy in vitro and in vivo in preclinical models of pancreatic cancer.

## 2. Results

### 2.1. Both PKM2 and LDHA are Expressed by Pancreatic Cancer Cells

All human pancreatic cancer cell lines analyzed expressed both PKM2 and LDHA at both proliferative and nonproliferative stages of cell culture. The BxPc-3, MIA PaCa-2, PK-45P, and PK-1 cell lines had a stable and relatively high expression of PKM2 and LDHA at both stages of cell culture in comparison to the other cancer cell lines analyzed (Figure 1). Since the BxPc-3 and MIA PaCa-2 cell lines had similar expression of both PKM2 and LDHA, we selected these two cell lines for further studies. The high expression of PKM2 observed in PDAC cells correlates with published data by Yokoyama et al. [7].

### 2.2. Synergistic Inhibition of Pancreatic Cancer Cell Proliferation with the Combined Treatment

The response of BxPc-3 and MIA PaCa-2 cells to TEPP-46 and FX-11 alone or in combination was assessed by the MTS assay. Treatment with TEPP-4 or FX-11 alone reduced cell metabolic activity in a concentration-dependent manner. The half maximum inhibitory concentration (IC_50_) of TEPP-46 single treatment was >75 µM for both cell lines. The IC_50_ of FX-11 was 49.27 µM for BxPc-3 and 60.54 µM for MIA PaCa-2 cells. When the two compounds were combined, a significant reduction in cell proliferation was observed compared with each single treatment in both cell lines (Figure 2A,B). The IC_50_ of the combination therapy was 30.3 µM and 40.2 µM for the BxPc-3 and MIA PaCa-2 cell lines, respectively. The combination treatment was considered synergistic as CI < 1 was observed for both cell lines: CI = 0.45 for BxPc-3 and CI = 0.48 for MIA PaCa-2.

PK and LDHA enzyme activity were evaluated in response to treatment with TEPP-46 and FX-11. There was a positive correlation between PK activity and increasing TEPP-46 concentrations and an inverse correlation between LDHA activity and increasing FX-11 concentrations in both cell lines (Figure 2C,D). Moreover, there was a significant inverse correlation between PK and LDHA activity for both cell lines when treated with increasing concentrations of TEPP-46 and FX-11 (Figure 2E,F).

BxPc-3 and MIA PaCa-2 cells were also stained with hematoxylin to microscopically assess the effect of each treatment on cell morphology and proliferation rate. There was a significant reduction in the size of the colonies and the number of viable cells with increasing concentrations of TEPP-46 and FX-11 compared with controls (0 µM group), and the combined treatment further reduced cell density, proliferation rate, and viability in both cell lines compared to TEPP-46 or FX-11 alone (Figure 2G,H).

### 2.3. Combination Therapy Significantly Attenuated Tumor Growth in the Subcutaneous Tumor Model

We assessed efficacy in vivo in a subcutaneous BxPc3-Luc tumor xenograft model. All treatments significantly reduced tumor growth compared with controls (Figure 3A–C). However, FX-11 and both low- and high-dose combination treatments significantly delayed tumor growth compared with TEPP-46 monotherapy, whereas the high-dose combination protocol significantly reduced tumor growth compared to all treatments (Figure 3A–C).

All treatments were well tolerated, as mice did not encounter any significant changes in body weight, liver enzyme function, and albumin compared with the control group, indicating no observable toxicity (Figure 3D,E).

### 2.4. Significant Therapeutic Efficacy with the Combination Therapy in the Orthotopic Tumor Model

We also assessed the effect of high-dose combination therapy in a more clinically relevant orthotopic tumor model, generated with BxPc3-Luc cells. Similar to the results from the subcutaneous tumor model, the combination treatment significantly delayed tumor growth (Figure 4A,B). At the end of the experiment, post-mortem examination revealed liver and spleen metastases in the control groups; however, no metastases were observed in the animals treated with the combination therapy. Tumor weights were significantly lower in the combination treatment group compared with controls (Figure 4B). No significant weight loss was encountered by the mice (Figure 4C).

### 2.5. PK and LDHA Enzyme Activity in Plasma and Tumor Lysates

PK enzyme activity was significantly higher in plasma and tumor lysates from the TEPP-46, low- and high-dose combination therapy groups compared with controls, whereas no significant differences were found between the FX-11 treated group and controls (Figure 5A). In contrast, LDHA activity was significantly lower in plasma and tumor lysates from the FX-11, low- and high-dose combination groups compared with controls, and no difference was observed between the TEPP-46 and control groups (Figure 5B). Overall, enzyme activity in plasma was reflected by enzyme activity in tumor lysates.

### 2.6. Reduced PKM2 Detection, LDHA Expression, and Significantly Decreased Proliferation in Tumor Tissues

To further assess response to treatment, the detection of inactive PKM2 (dimeric form), and the expression of LDHA and Ki-67, was examined in tumor sections from each treatment group by immunohistochemistry. PKM2 detection and LDHA expression were considerably lower in tumor sections from all treatment groups compared with controls (Figure 6A).

The expression of the proliferation marker Ki-67 was significantly lower in the FX-11, low- and high-dose combination treatment groups compared with controls, whereas there were no significant differences between the TEPP-46 and control groups (Figure 6A,B). Moreover, proliferation indices were significantly lower in the high-dose combination group compared with all other treatments (Figure 6A,B). Similarly, significantly decreased proliferation indices were observed in tumor sections from the high-dose combination therapy compared with controls from the orthotopic tumor model study (Figure 6C).

## 3. Discussion

Altered energy metabolism is a hallmark of cancer and plays a crucial role in tumor initiation, progression, and resistance to treatment [4,6,13,26]. Cancer cells can use aerobic glycolysis to divert glucose metabolites from ATP production toward the synthesis of cellular building blocks (nucleotides, amino acids, and lipids) to meet the demands for sustained proliferation. This shift in cellular metabolism can be achieved through increased expression of glycolytic enzymes, such as PKM2 and LDHA. These enzymes are overexpressed in various malignancies, including pancreatic cancer, and we and others have shown that this is associated with poor prognosis [32,34,35,37,38,39].

In this context, we examined whether targeting PKM2 and LDHA can alter tumor cell metabolism and inhibit tumor cell proliferation. We demonstrate that the PKM2 activator, TEPP-46, and the LDHA inhibitor, FX-11, can effectively activate PKM2 and inhibit LDHA activity, respectively, both in vitro and in vivo. We also show that combining TEPP-46 and FX-11 can significantly and synergistically inhibit pancreatic cancer cell proliferation in vitro, and significantly delay tumor growth in vivo in preclinical models of pancreatic cancer, without observable toxicity. To our knowledge, our study is the first to examine this combination strategy in pancreatic cancer.

Our in vitro studies demonstrate that pancreatic cancer cells express both PKM2 and LDHA and that the activity of both enzymes can be effectively altered by TEPP-46 and FX-11. PKM2 activity was increased with increasing concentrations of TEPP-46 and LDHA activity was decreased with increasing concentrations of FX-11. Moreover, there was a significant inverse correlation between the activity of both enzymes. Cell metabolic activity decreased in a concentration-dependent manner with TEPP-46 and FX-11, and the combination of both had synergistic effects on reducing pancreatic cancer cell proliferation.

Similar to our in vitro results, both agents significantly reduced tumor growth in vivo compared with control groups. Indeed, these findings are in agreement with other studies reporting therapeutic activity with single agent TEPP-46 or FX-11 in various tumor models, including pancreatic cancer [13,24]. However, single-agent FX-11 and both low- and high-dose combination treatments were significantly more effective than TEPP-46 alone, whereas the high-dose combination therapy resulted in the best response. The significant delay in tumor growth was accompanied by increased PK activity and decreased LDHA activity in plasma and tumor lysates, and reduced detection of PKM2 and expression of LDHA in tumor tissues, along with significantly decreased tumor proliferation. The decreased detection of PKM2 in tumor sections is reflective of enzyme activation as a result of TEPP-46, which binds to a pocket at the PKM2 subunit interface, enhancing the association of PKM2 subunits into stable tetramers [13] (tetramer form not detected by PKM2 antibody used for immunohistochemistry. This mechanism of tetramer stabilization by TEPP-46 is reported to be refractory to inhibition by tyrosine-phosphorylated proteins and to have an impact on cell metabolism [13].

Although TEPP-46 can enhance PKM2 activity, leading to reduced lactate production and lipid synthesis, treatment with TEPP-46 alone does not induce cytotoxic changes and it is likely that cancer cells persist in a quiescent state, which has been linked to treatment resistance and disease recurrence. The metabolic requirements of different cells vary, and while tumor cells might adopt various means of gaining energy, decreased PKM2 activity might not be required for the survival of all tumor cells.

In addition to its function as a glycolytic enzyme in the cytoplasm, the dimeric form of PKM2 can also act as a protein kinase and a transcriptional co-activator in the nucleus. PKM2 can promote cell cycle progression by regulating mitotic checkpoint, chromosome segregation, and cytokinesis [7,40,41,42]. Nuclear PKM2 can promote the transcriptional activities of hypoxia-inducible factor (HIF), β-catenin, STAT 3, and Oct 4 [43,44,45,46]. β-catenin-mediated transactivation of cyclin D and c-Myc can promote the expression of Cdc25A, GLUT1, LDHA, and PKM2 [47], while HIF-1α can induce transcription of LDHA, PDK1, and VEGF. The upregulated expression of glycolytic genes enhances the Warburg effect, while cyclin D expression promotes G1-S phase transition, collectively encouraging cell proliferation and tumor progression.

It is likely that in addition to the depletion of glycolytic intermediates, tetramerization of PKM2 by TEPP-46 might also impede nuclear translocation of PKM2 and hence inhibit transcription of LDHA, possibly contributing to the effectiveness of the combination treatment. Inhibition of LDHA by FX-11 can reduce the ability of cancer cells to metabolize pyruvate to lactate, halting the regeneration of NAD+, which is required for sustained glycolysis, and consequently prevent tumor cell proliferation [24]. Inhibition of LDHA has been reported to impair lymphoma and pancreatic cancer growth by decreasing ATP levels, inducing oxidative stress and cell death [24].

Both TEPP-46 and FX-11 alone reduced tumor growth and decreased tumor cell proliferation. However, combined treatment using the high-dose regimen of TEPP-46 and FX-11 significantly enhanced efficacy and produced synergistic antitumor activity without apparent toxicity—although a limitation of the study is that the in vivo studies were performed using only one pancreatic cancer cell line. However, in phase I–III trials, metabolism modulating agents have generally also been well tolerated.

In summary, we demonstrate that TEPP-46 and FX-11 can be safely and effectively combined to synergistically reduce pancreatic tumor growth. Although most inhibitors are still in the preclinical phase, the targeting of glycolytic enzymes, such as PKM2 and LDHA, represents a very promising approach for the treatment of pancreatic cancer.

## 4. Materials and Methods

### 4.1. Cell Lines and Chemicals

Nine different human pancreatic cancer cell lines were purchased from RIKEN BioResource Centre (RIKEN BRC, Tsukuba, Japan) and PerkinElmer (Caliper LifeSciences, Hopkinton, MA, USA). PANC-1, PK-1, PK-45H, KLM-1, BxPc-3, PK45P, and PK-59 were maintained in RPMI-1640 medium, while MIA PaCa-2 was maintained in DMEM and KP-4 in DMEM/F12 medium. All media were supplemented with 10% fetal bovine serum (FBS), 1% penicillin/streptomycin, and 2mM glutamine. Cells were maintained in a humidified atmosphere of 21% O_2_, 5% CO_2_ at 37 °C.

The PKM2 activator IV, TEPP-46, was obtained from Cayman Chemicals (Ann Arbor, MI, USA) and the lactate dehydrogenase A inhibitor, FX-11, was obtained from Toronto Research Chemicals Inc. (Toronto, ON, Canada).

### 4.2. Expression of PKM2 and LDHA in Pancreatic Cancer Cells

The expression levels of PKM2 and LDHA were evaluated in the abovementioned pancreatic cancer cell lines by western blotting when cells reached approximately 50% and 90% of confluence in culture (proliferative and nonproliferative stage, respectively).

Whole protein content was extracted by lysis using RIPA buffer (Sigma-Aldrich, St Louis, MA, USA) complemented with a tablet of complete EDTA-free protease inhibitors (Roche Diagnostics Ltd., Welwyn Garden City, UK). Total protein concentration was measured by the BCA assay (Pierce, Rockford, IL, USA). Then 15 µg protein was run on pre-cast gels (NuPAGE Novex 4–12% Bis-Tris 1.0 mm, 10 wells gel, Invitrogen, Waltham, MA, USA) and transferred onto 0.45 µm pore size polyvinylidene difluoride (PVDF) membrane (Invitrogen). The membrane was blocked with a 5% BSA solution and incubated overnight at 4 °C with mouse anti-PKM2 (DF-4, ScheBo^®^Biotech, Giessen, Germany) or rabbit anti-LDHA antibodies (Cell Signaling, London, UK) (1:1000 dilution), followed by incubation with appropriate horseradish peroxidase conjugated secondary antibodies (Cell Signaling; 1:2000). The antigen–antibody reaction was detected using an enhanced chemiluminescence substrate (Thermo Scientific, Waltham, MA, USA), and anti-β-actin antibody (Cell Signaling; 1:1000) was used as a protein loading control. Relative protein expression was evaluated as chemiluminescence intensity of PKM2 and LDHA normalized to chemiluminescence intensity of β-actin. Densitometry readings and analysis were performed using the ImageJ software and have been included as Appendix A.

### 4.3. Evaluation of Cell Proliferation Rate

The MTS assay was used to assess the effect of treatments on tumor cell proliferation (Abcam, Cambridge, UK). Briefly, pancreatic cancer cells were seeded at a density of 3000–5000 cells per well in 96-well plates. Cells were treated with TEPP-46, FX-11, or a combination of both agents for 72 h and cell proliferation was measured as per manufacturer’s protocol. To evaluate whether the combination treatment was synergistic, the combination index (CI) was calculated based on the Chou and Talalay method from dose-response data (CI = 1, additive effect; CI < 1, synergism; CI > 1, antagonism) [48]. The CompuSyn software (ComboSyn, Inc., Paramus, NJ, USA) was used for drug dose–response analysis and calculation of the dose–effect curve, IC_50_, and combination index (CI). The algorithm is based on the same Chou-Talalay’s Combination Index Theorem that CalcuSyn is based on:$$\mathrm{CI}=\frac{{\left(D\right)}_{1}}{{\left(Dx\right)}_{1}}+\frac{{\left(D\right)}_{2}}{{\left(Dx\right)}_{2}}$$
where (*D_x_*)_1_ = dose of drug 1 to produce 50% cell kill alone; (*D*)_1_ = dose of drug 1 to produce 50% cell kill in combination with (*D*)_2_; (*D_x_*)_2_ = dose of drug 2 to produce 50% cell kill alone; (*D*)_2_ = dose of drug 2 to produce 50% cell kill in combination with (*D*)_1_.

### 4.4. In Vitro PK and LDHA Enzyme Activity

PK and LDHA enzyme activity were assessed in response to TEPP-46 and FX-11 by commercially available PK and LDHA activity assay kits (BioVision, Milpitas, CA, USA) as per manufacturer’s protocols. Briefly, pancreatic cancer cells were seeded in 6 well-plates and treated with different concentrations of either TEPP-46 or FX11 (0–100 µM) for 6 h, following which, PK and LDHA activity was measured and compared with controls.

### 4.5. Tumor Models and Assessment of In Vivo Efficacy and Toxicity

Immunocompromised CD-1 mice (age, 6–8 weeks; weight, 20–25 g) were purchased from Charles River Laboratories Inc., (Harlow, UK). All animal experiments were performed in accordance with the UK Home Office Animals Scientific Procedures Act 1986 and UK Co-ordinating Committee on Cancer Research Guidelines for the Welfare and Use of Animals in Cancer Research [49] and approved by the University College London Animal Welfare and Ethical Review Body (UK Home Office approved Animal Project License PPL PPL-70/7100, 2013).

Subcutaneous xenografts were established using the luciferase-expressing human pancreatic cancer cell line BxPc-3. Briefly, 6 × 10^6^ cells were injected into the right flank of mice and allowed to grow until reaching a volume of ~150–200 mm^3^. Mice (*n* = 5 per group) with similar tumor volumes were allocated into the following groups: (1) Control (40% (2-hydroxypropyl) beta-cyclodextrin in saline), (2) TEPP-46 (30 mg/kg), (3) FX-11 (2 mg/kg), (4) Low-dose combination therapy of TEPP-46 and FX-11 (15 mg/kg and 1 mg/kg, respectively), and (5) High-dose combination therapy of TEPP-46 and FX-11 (30 mg/kg and 2 mg/kg, respectively). TEPP-46 and FX11 were prepared in 40% (2-hydroxypropyl) beta-cyclodextrin solvent in saline. Daily injections (100 µL) of the vehicle or treatments were administered intraperitoneally for a period of 3 weeks. Tumor volumes were measured thrice a week using a digital caliper and calculated using the following formula: Tumor Volume = length (mm) × width (mm) × width (mm) × 0.52.

Mice were weighed three times per week, and toxicity was evaluated as changes in body weight, liver enzyme function [alanine transaminase (ALT), aspartate transaminase (AST)], and albumin (Roche Diagnostics Ltd.). Mice were sacrificed following the three-week treatment period, approximately one hour after the last injection; blood samples were collected via cardiac puncture and tumors were harvested. Blood was collected in heparinized polystyrene tubes and the plasma was immediately separated by centrifugation at 10,000 rpm for 15 min at 4 °C; plasma was stored at −80 °C until analysis. Tumors were weighed, and half of each tumor tissue sample was flash-frozen in liquid nitrogen or fixed in formalin.

### 4.6. Orthotopic Tumor Model

In order to enhance the clinical relevance of this study, orthotopic tumor models were generated in immunocompromised CD-1 mice by injecting BxPc3-Luc cells directly into the pancreas, using a well-established method [50]. Briefly, mice were anesthetized with 2–3% isoflurane, and an incision was made in the abdominal cavity directly above the pancreas to allow visualization of the pancreatic lobes. The pancreas was gently retracted and injected with 6 × 10^6^ BxPc3-Luc cells in a 100 µL solution of 1:1 PBS and matrigel. A dissection microscope was used to confirm delivery of the tumor cells under the capsule of the pancreas. The pancreas was then placed back within the abdominal cavity and both muscle and skin layers were sutured. All procedures were conducted under sterile conditions. Tumor engraftment was confirmed by bioluminescence imaging using the IVIS system (PerkinElmer/Caliper Life Sciences, Hopkinton, MA, USA) as previously described [51], and mice were allocated to 2 groups with similar tumor burden based on bioluminescence signal (*n* = 5 per group): (1) Vehicle control, and (2) High-dose combination therapy of TEPP-46 and FX-11, as described above. Mice were monitored and weighed thrice weekly. At the end of the treatment period, mice were imaged once more prior to sacrifice and tumors were harvested and weighed.

### 4.7. PK and LDHA Activity in Plasma and Tumor Lysates

The enzymatic activity of PK and LDHA was evaluated in tumor lysates and plasma samples from each treatment group from the subcutaneous tumor model studies, using a commercially available kit (BioVision) as per manufacturer’s protocol.

### 4.8. Immunohistochemistry

Excised tumors were formalin-fixed, paraffin-embedded, and sectioned. Tumor sections from each treatment group (from the subcutaneous tumor model) were stained with hematoxylin and eosin, and the expression of dimeric PKM2 (inactive), LDHA, and the proliferation marker Ki-67 was evaluated by immunohistochemistry. Samples from the orthotopic tumor model were stained for Ki-67 only.

Briefly, sections were subjected to heat-mediated antigen retrieval and endogenous peroxidase activity was blocked using 3% hydrogen peroxide for 20 min, followed by incubation with 2.5% normal horse blocking serum for 20 min. Sections were then incubated with primary antibody for 1 h at room temperature for Ki-67 (1:300, Dako, Didcot, UK), and overnight at 4 °C for PKM2 (1:100, ScheBO^®^Biotech, Giessen, Germany) and LDHA (1:250, ScheBO^®^Biotech). Immunoreactivity was visualized using horseradish peroxidase (HRP)—Conjugated secondary antibodies (Vector Laboratories, Peterborough, UK) and 3, 3′-diaminobenzidine (DAB; Dako). Sections were then counterstained with hematoxylin, dehydrated, and mounted with DPX (Sigma Aldrich, Dorset, UK). Proliferation indices were evaluated by counting positively stained nuclei in 2–3 random regions and expressed as percentage of positively stained nuclei per 1000 tumor cells using a standardized grid.

### 4.9. Statistical Analyses

Data were plotted and analyzed using Prism (GraphPad Software, USA). Two-way ANOVA and multiple t-tests were used to assess differences between groups, and linear and nonlinear regression analyses were used for correlation of results. Results were considered statistically significant at *p* < 0.05. The CompuSyn software (ComboSyn, Inc., Paramus, NJ, USA) was used for analyses of synergy and for calculation of combination indices (CIs). If the CI = 1, the effect was considered additive, if the CI < 1, the combination effect was considered synergistic, and if the CI > 1, the effect was considered antagonistic.

## 5. Conclusions

In this study, we investigated whether the PKM2 activator, TEPP-46, and the LDHA inhibitor, FX-11, can be effectively combined to inhibit in vitro and in vivo tumor growth in preclinical models of pancreatic cancer. Our results demonstrate that combining TEPP-46 with FX-11 synergistically inhibited pancreatic cancer cell proliferation and significantly delayed tumor growth in vivo compared with all other groups, without apparent toxicity. Dual treatment with TEPP-46 and FX-11 resulted in increased PK and reduced LDHA enzyme activity in plasma and tumor tissues and decreased PKM2 and LDHA expression in tumors, which was reflected by a decrease in tumor volume and proliferation. Although most inhibitors are still in the preclinical phase, the targeting of glycolytic enzymes, such as PKM2 and LDHA, represents a very promising therapeutic approach for the treatment of pancreatic cancer.

## Figures and Tables

**Figure 1 cancers-11-01372-f001:**
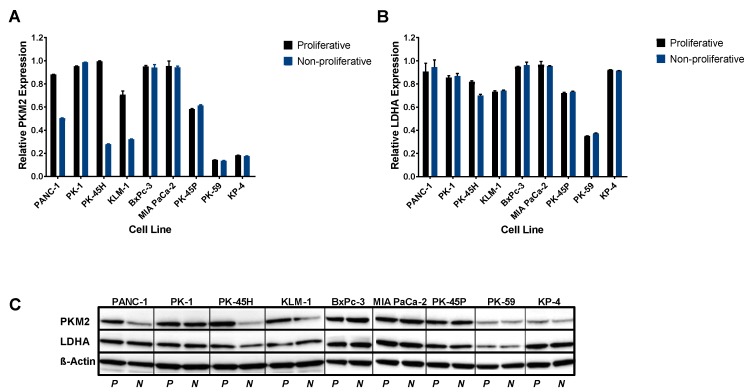
Pyruvate kinase M2 (PKM2) and lactate dehydrogenase A (LDHA) expression in human pancreatic cancer cell lines. (**A**) Relative PKM2 expression and (**B**) Relative LDHA expression in proliferative and nonproliferative phases of cell culture in nine pancreatic cancer cell lines. (**C**) Western blots showing expression levels compared with loading controls across the cell lines. Data presented as relative expression ± SE, obtained from chemiluminescence intensity of PKM2 and LDHA normalized to chemiluminescence intensity of β-actin.

**Figure 2 cancers-11-01372-f002:**
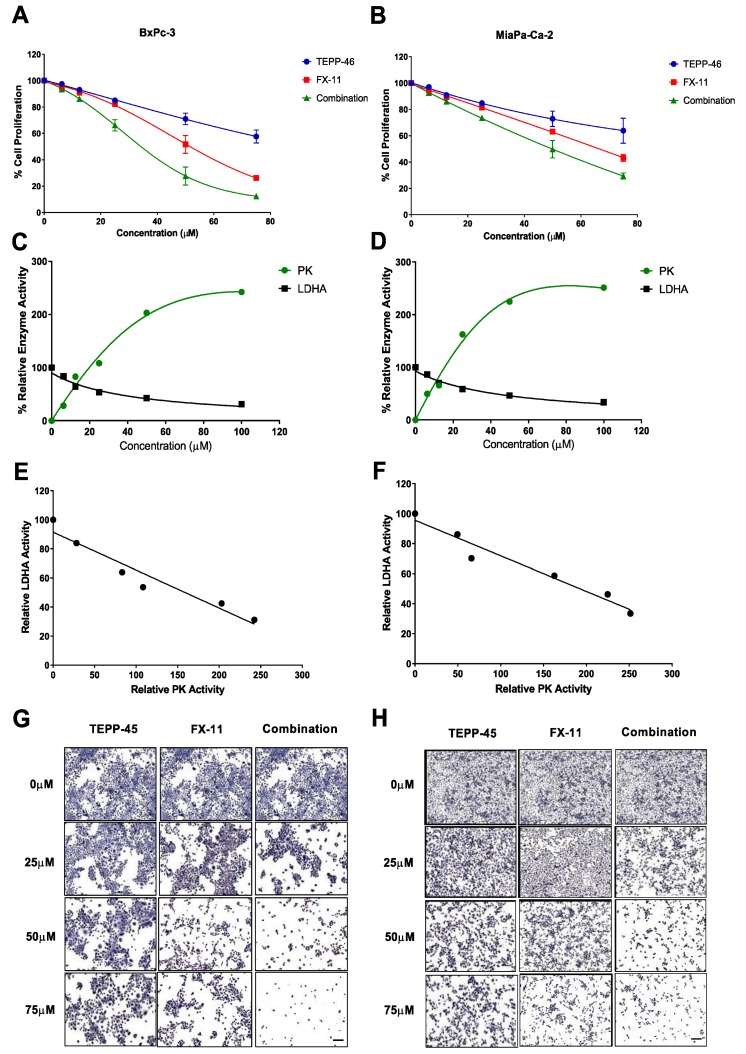
Effect of TEPP-46 and FX-11 on BxPc-3 and MIA PaCa-2 cells. Cell proliferation in response to increasing concentrations of TEPP-46 and FX-11 alone or in combination in (**A**) BxPc-3 cells and (**B**) MIA PaCa-2 cells. Relative PK and LDHA enzyme activity with increasing concentrations of TEPP-46 and FX-11 in (**C**) BxPc-3 cells and in (**D**) MIA PaCa-2 cells. There was a correlation between PK activity and increasing TEPP-46 concentrations (R^2^ = 0.98 and R^2^ = 0.97 for BxPc-3 and MIA PaCa-2 cells, respectively) and between LDHA activity and increasing concentrations of FX-11 (R^2^ = 0.94 and R^2^ = 0.96 for BxPc-3 and MIA PaCa-2 cells, respectively). PK activity increased with increasing concentrations of TEPP-46, and LDHA activity decreased with increasing concentrations of FX-11. There was a significant inverse correlation between PK and LDHA enzyme activity with increasing concentrations of TEPP-46 and FX-11 in both cell lines (**E**) BxPc-3 (R^2^ = 0.93, *p* = 0.002) and (**F**) MIA PaCa-2 (R^2^ = 0.95, *p* = 0.001). Decreased cell proliferation with increasing concentrations of TEPP-46, FX-11, and a combination of both was confirmed by hematoxylin staining in (**G**) BxPc-3 and (**H**) MIA PaCa-2 cells. Scale bar: 500 µm. Cell proliferation data presented as mean ± SE; means plotted for enzyme activity correlations.

**Figure 3 cancers-11-01372-f003:**
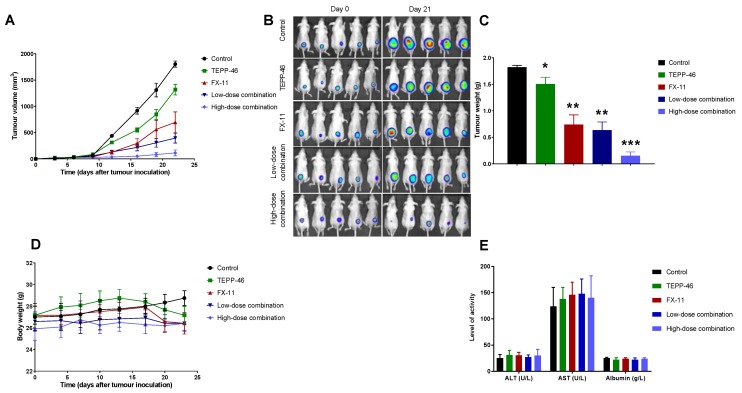
Efficacy and toxicity evaluation of TEPP-46, FX-11, and combination therapy in the subcutaneous BxPc-3-Luc tumor model. Efficacy was evaluated based on (**A**) Tumor volume over time, (**B**) Bioluminescent images of mice from each treatment group at the start and end of treatment (day 0 and day 21, respectively), and (**C**) Tumor weights at the end of treatment. Toxicity was evaluated based on change in (**D**) Weight of mice over the course of therapy and (**E**) Liver enzyme function and albumin. Each treatment significantly delayed tumor growth compared with the control group; FX-11, low- and high-dose combination therapy significantly reduced tumor growth compared with TEPP-46, and the high-dose combination therapy significantly reduced tumor growth compared with all other treatments (*p* < 0.05, 2-way ANOVA and multiple t-tests). No significant weight loss or change in liver enzyme function and albumin were encountered in the treatment groups compared with control mice. Data presented as mean ± SE; * indicates significantly different from controls; ** indicates significantly different from controls and TEPP-46; *** indicates significantly different from controls and all other treatment groups.

**Figure 4 cancers-11-01372-f004:**
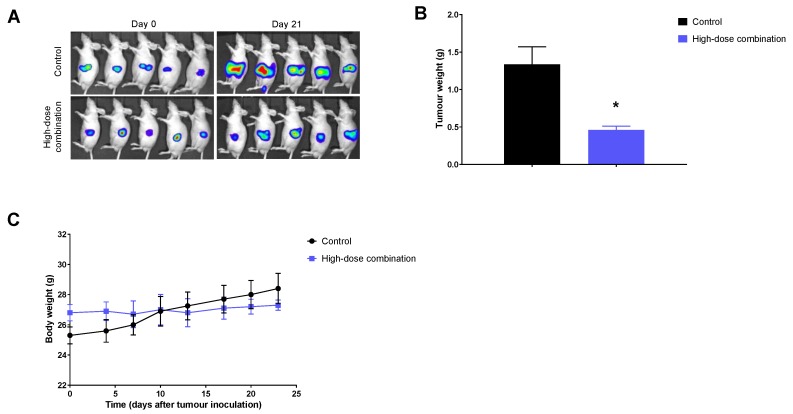
Efficacy and toxicity evaluation of the combination therapy in the BxPc-3-Luc orthotopic tumor model. (**A**) Bioluminescent images of mice from the control and high-dose combination therapy group obtained at the start and end of treatment (day 0 and day 21, respectively); (**B**) Tumor weights at the end of treatment and (**C**) Weight of mice over the course of therapy. The combination therapy significantly reduced tumor growth compared with the control group (*p* = 0.006, Student’s *t*-test). No significant changes in weight were encountered by the treated mice compared with the controls. Data presented as mean ± SE; * indicates significantly different from controls.

**Figure 5 cancers-11-01372-f005:**
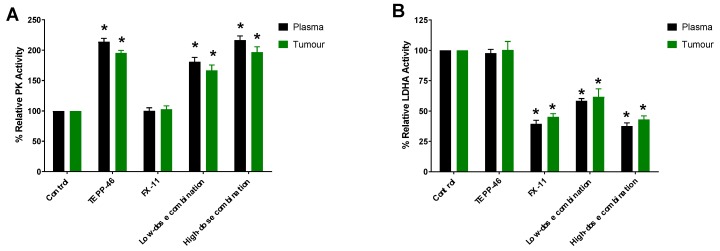
Glycolytic enzyme activity in plasma and tumor lysates in response to TEPP-46, FX-11, and combination therapy in the subcutaneous BxPc-3-Luc tumor model. (**A**) PK activity and (**B**) LDHA activity. Overall, TEPP-46 treatment increased PK activity and FX-11 treatment decreased LDHA activity in plasma and tumor lysates. Enzyme activity in plasma was reflected by enzyme activity in tumor lysates. Data presented as mean ± SE. * indicates significantly different from controls (*p* < 0.05, 1-way ANOVA).

**Figure 6 cancers-11-01372-f006:**
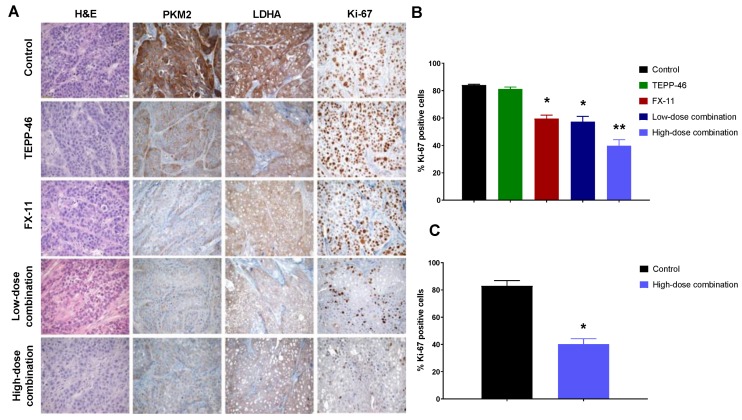
Detection of inactive PKM2 and LDHA and Ki-67 expression in response to TEPP-46, FX-11, and combination treatment. (**A**) Representative images of tumor sections from each group from the subcutaneous tumor model stained with H&E, PKM2, LDHA, and Ki-67 (40× magnification). Scale bar: 20 µm; Proliferation indices in response to treatment in the (**B**) subcutaneous and (**C**) orthotopic tumor model. PKM2 detection and LDHA expression were lower in tumor sections from all treatment groups compared with controls. Ki-67 expression was significantly lower in the FX-11, low- and high-dose combination treatment groups compared with controls, and significantly lower in the high-dose combination group compared with all other treatments. Significantly decreased proliferation indices were observed in tumor sections from the high-dose combination treatment compared with controls in the orthotopic tumor model. Data presented as mean ± SE. * indicates significantly different from controls; ** indicates significantly different from controls and all other treatment groups.

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
