# Peer review of "Targeting Pyruvate Kinase M2 and Lactate Dehydrogenase A Is an Effective Combination Strategy for the Treatment of Pancreatic Cancer"

_cancers, 2019, doi:10.3390/cancers11091372_

Round 1
Reviewer 1 Report
This study by Mohammad et al. examines the therapeutic effect of (simultaneous) targeting two glycolysis-associated enzymes, namely PKM2 and LDHA, for the treatment of pancreatic ductal adenocarcinoma.
Experimental planning and sequence, reported data and the analyses appear thorough and solid, the structure and language of the manuscript are adequate. The manuscript totally fits the scope of Cancers, has merit, is of interest for the field and might be considered for publication, as soon as, from my point of view, the following few issues have been addressed:
Major:
The viability assay the authors use for measurement of cell proliferation is a metabolic (!!!) reduction-assay and depends on the presence of NAD(P)H. In the context of therapeutic modulation of glycolysis I have to insist on repeating these experiments with a non-metabolic assay. The conclusions drawn in the discussion section in lines 286-288 are not supported by the data provided. If the authors insist on keeping this part of the paragraph they need to include/add on data from e.g.: Seahorse analysis (in vitro) Response to 2DG-treatment (in vitro) Oxidative stress measures (in vitro + in vivo) Cell death analyses (in vitro + in vivo)
Minor:
Experiments have been performed with only two (in vitro) or even one (in vivo) human cell line. Broader confirmation of the results would require additional cell lines, especially in the in vivo.This fact should at least be addressed and acknowledged as limitation of the study – e.g. in the discussion section. The in vitro doses required for the demonstrated effects are in high micro-molar ranges – this usually suggests/predicts low specificity/potency and in vivo toxicity of some sort. The authors present weight and liver fct. tests as tox-readout, which I feel is adequate in this setting. However, I would urge the authors to speculate on possible side-effects in a perspectively conducted human trial with these drugs and their combination. In the results section the terminology regarding measuring PKM2 expression by IHC is misleading for the reader. The mechanism of multimerization and consequently lack of detection is perfectly explained in the discussion section. I would like to see this mentioned earlier on, to avoid confusion. > Here I suggest using the term ‚PKM2 detection’ rather than ‚ PKM2 expression’
Reviewer 2 Report
The manuscript entitled “Targeting pyruvate kinase M2 and lactate dehydrogenase A is an effective combination strategy for the treatment of pancreatic cancer” by Goran Hamid Mohammad et al., demonstrated the in vitro and in vivo combination effects of TEPP-46 and FX-11 in pancreatic cancer. The manuscript needs to be improved upon consideration of the following points:
Major
Line 121-122, For the description and calculation of combination index (CI) (at IC50 level?) based on dose-response curves in Figure 2, provide the IC50 values of TEPP-46 and FX-11 single treatment in the result section. Also, briefly provide the CI formula in the materials and methods for the journal readers. There is no matching between fig.2, result section, and its legend. Even though xenograft models indicate significantly delayed tumor growth in a high-dose combination group (Fig. 3) compared with a single treatment group, there is no single treatment group for the comparison of the efficacy of combination of both agents in orthotopic tumor models in Fig. 4. It is not sufficient to support the main results that combination of both agents is effective in the treatment of pancreatic cancer. Regarding LDHA activity and expression after treatment of TEPP-46, TEPP-46 treatment does not affect the activity of LDHA in Fig. 5, but there is a significant reduction of LDHA expression in Fig. 6 and its description. What can drive this discrepancy? What is a cause of difference between Ki-67 (Fig. 6B) and tumor weight (Fig. 3C) for the treatment of TEPP-46 single treatment?
Minor
Authors argued that the high expression of PKM2 was observed in PDAC cells (line 107). However, there is no control (e.g., normal pancreas cell lines) to compare the levels of PKM2. In addition, it is hard to say without any indicators that cells reached at 90% confluence are non-proliferating cells. There is no Y-axis title in Fig. 3E. Line 81: Delete a square bracket Line 170: typo, form Line 351: typo, thrice Add scale bars in Fig. 2E, F and Fig. 6A
Round 2
Reviewer 1 Report
All comments and concerns have been adequately addressed and discussed in the rebuttal letter as well as in the revised manuscript.
I won't insist on the proposed additional experiments if the editorial board and other reviewers agree to accept the revised manuscript in its present form.
Reviewer 2 Report
In this revised version of manuscript, Mohammad and colleagues have improved the manuscript in an impressively short time. All required experiments have been included and the confusion about some structure descriptions has been resolved with appropriate terminology. I do not have any further concerns.